# Establishment of an Antiplasmodial Vaccine Based on PfRH5-Encoding RNA Replicons Stabilized by Cationic Liposomes

**DOI:** 10.3390/pharmaceutics15041223

**Published:** 2023-04-12

**Authors:** Wesley L. Fotoran, Jamile Ramos da Silva, Christiane Glitz, Luís Carlos de Souza Ferreira, Gerhard Wunderlich

**Affiliations:** 1Department of Parasitology, Institute for Biomedical Sciences, University of São Paulo, Avenida Professor Lineu Prestes 1374, São Paulo 05508-000, SP, Brazil; 2Department of Microbiology, Institute for Biomedical Sciences, University of São Paulo, Avenida Professor Lineu Prestes 1374, São Paulo 05508-000, SP, Brazil; 3Department of Molecular Physiology, Institute of Animal Physiology, Westfälische Wilhelms University of Münster, 48149 Münster, Germany; 4Scientific Platform Pasteur-USP, University of São Paulo, Avenida Lucio Martins Rodrigues 370, São Paulo 05508-020, SP, Brazil

**Keywords:** nucleic acid vaccines, RNA replicons, cationic liposomes, malaria, PfRH5, intradermal immunization, tattooing

## Abstract

Background: Nucleic acid-based vaccines have been studied for the past four decades, but the approval of the first messenger RNA (mRNA) vaccines during the COVID-19 pandemic opened renewed perspectives for the development of similar vaccines against different infectious diseases. Presently available mRNA vaccines are based on non-replicative mRNA, which contains modified nucleosides encased in lipid vesicles, allowing for entry into the host cell cytoplasm, and reducing inflammatory reactions. An alternative immunization strategy employs self-amplifying mRNA (samRNA) derived from alphaviruses, but lacks viral structural genes. Once incorporated into ionizable lipid shells, these vaccines lead to enhanced gene expression, and lower mRNA doses are required to induce protective immune responses. In the present study, we tested a samRNA vaccine formulation based on the SP6 Venezuelan equine encephalitis (VEE) vector incorporated into cationic liposomes (dimethyldioctadecyl ammonium bromide and a cholesterol derivative). Three vaccines were generated that encoded two reporter genes (GFP and nanoLuc) and the *Plasmodium falciparum* reticulocyte binding protein homologue 5 (PfRH5). Methods: Transfection assays were performed using Vero and HEK293T cells, and the mice were immunized via the intradermal route using a tattooing device. Results: The liposome–replicon complexes showed high transfection efficiencies with in vitro cultured cells, whereas tattooing immunization with GFP-encoding replicons demonstrated gene expression in mouse skin up to 48 h after immunization. Mice immunized with liposomal PfRH5-encoding RNA replicons elicited antibodies that recognized the native protein expressed in *P. falciparum* schizont extracts, and inhibited the growth of the parasite in vitro. Conclusion: Intradermal delivery of cationic lipid-encapsulated samRNA constructs is a feasible approach for developing future malaria vaccines.

## 1. Introduction

The development of nucleic acid vaccines began not only as an attempt to decrease the dependency on vaccines with live or attenuated vectors, but also as an investigation of the effects of the direct injection of DNA or RNA expression vectors [1]. Plasmid DNA-based vaccines are still the most widely studied form of nucleic acid vaccine approaches, and immunization with purified circular plasmid DNA has already been shown to be successful in various tissues of small animals, even when deploying different routes of administration [1,2,3,4]. Despite being a flexible platform and successful in small animals tested in laboratories, DNA vaccines were not approved for human use until the pandemic scenario of SARS-CoV-2 in 2021, one year after the approval of mRNA-based vaccines in 2020. This delay has been ascribed to concerns regarding the potential oncogenic effects of DNA integration, and because DNA vaccines have shown lower potency in human clinical trials [5]. Among the various types of RNA vaccines, only mRNA vaccines are currently approved; however, self-assembly/amplifying (sam) vaccines represent a promising immunogenic approach to nucleic acid vaccines. Replicase-containing RNA vectors have been shown to significantly more immunogenic than the conventional plasmid DNA, and recent results have shown that a single dose of 0.1 µg of these vectors injected intramuscularly was able to immunize mice [6]. This methodology uses “replicons”: self-amplifying RNA vectors (samRNA) that lack a viral capsid and envelope, but contain the nonstructural protein genes that encode a viral replicase, 5′ and 3′ sequences important for replication, and a subgenomic promoter derived from alphavirus vectors [7]. An increased number of RNA copies also leads to enhanced levels of transient gene expression, both in vitro and in vivo [8]. Replicon particles derived from Venezuelan equine encephalitis (VEE) are an efficient platform for RNA vaccines, with robust effects on vaccine formulations when complexed with lipid formulations, promoting the protection of RNA from degradation and enhancing transfection efficiency [9].

Liposomes are artificial round lipid-based nanoparticles consisting of one or more phospholipid bilayer membranes that encapsulate an internal aqueous compartment [10]. They have played a major role as model systems for membrane studies; however, more recently, they have been successfully used as carriers for drugs and vaccines [11]. Depending on the modification of the lipid moieties with cationic or anionic side groups, formed liposomes gain different affinities for charged or uncharged cargo molecules. These may be attached to the inner or outer surface of liposomes or confined inside the liposomes, protecting the cargo from degradation; this is important when the cargo molecules are nucleic acids (revised in [12]). The DDAB/DC-Chol formulation used here is based on previously published studies on pDNA vaccines [13,14]. The high stability of DDAB/DC-Chol liposomes in association with pDNA has been previously reported for the development of antimalarial DNA vaccines [14]; however, to date, the same approach has not been tested for mRNA delivery.

In this study, RNA replicons containing reporter genes that encode green fluorescent protein (GFP) and nanoluciferase (nLuc) incorporated into cationic liposomes were used for the in vitro transfection of Vero and HEK 293T cells, and for intradermal immunization of BALB/c and C57BL/6 mice by tattooing. The samRNA used in this study was derived from VEE vectors, and synthesized from a plasmid DNA template using the DNA-dependent SP6 RNA polymerase. Constructs containing the SP6 promoter, specifically SP6-VEE-GFP and SP6-VEE-nLuc, were used for in vitro transcription, resulting in samRNAs with a high degree of purity. These RNA replicons—expressing either GFP or nLuc—were incorporated into DDAB/DC-Chol liposomes by sonication, and the resulting lipoplexes were used for in vitro and in vivo assays. Lipoplex formulations were used for the intradermal immunization of BALB/c and C57BL/6 mice.

To analyze the efficiency of the immunization assays, an intravital imaging system was used to measure the relative fluorescence or bioluminescence in the tattooed areas. The lipoplex-samRNA approach was applied to develop a potential vaccine against *Plasmodium falciparum* based on the vaccine candidate gene PfRH5, which is an essential and almost invariant ligand of the human blood cell receptor and a key for erythrocyte invasion of malarial parasites [15]. The results demonstrate that the samRNA-encoding PfRH5 represents a promising tool for the development of a safe and effective antimalarial vaccine delivered via the intradermal route.

## 2. Materials and Methods

### 2.1. Cloning of Plasmid Constructs

The plasmids used as templates for RNA replicon synthesis were SP6-VEE-IRES-Puro (ADDGENE: Plasmid #58971) [16], modified by adding the protein of interest, nanoluciferase, as an amplified PCR product, and SP6-VEE-GFP (ADDGENE: Plasmid #58976). The self-amplifying RNA (samRNA) of SP6-VEE-nLuc and SP6-VEE-PfRH5, the products of subcloning with the enzymes NotI and NdeI, are shown in Figure 1, along with the other constructs. For comparison, the following pcDNA3-derivates were used in the transfection and immunization assays: pcDNA3-GFP as a control for RNA replicons expressing GFP, and pcDNA3-Luc as a control for RNA replicons expressing nLuc. The plasmids were transformed into *E. coli* DH10B competent cells using standard molecular techniques, and produced on a large scale using the Plasmidprep protocol for high purity [17]. Additionally, for immunizations, stable, non-replicating mRNA encoding the PfRH5 antigen was used as a comparator. The PfRH5 gene was cloned into the vector TEV-MCS-101A kindly provided by Dr. Katalin Karikó (BioNTech, Mainz, Germany). In this vector, the mRNA for PfRH5 is flanked in the 5′ by the Tobacco etch virus (TEV) 5′ untranslated region (UTR), and in the 3′ by Xenopus beta-globin 3′UTR followed by 101 adenines.

### 2.2. RNA Replicon In Vitro Synthesis

The DNA plasmids SP6-VEE-GFP and SP6-VEE-nLuc were linearized downstream of the 3′ end of the RNA replicon template sequence by digestion with MluI. Linearized DNA templates were transcribed into RNA using a kit from RiboMAX™ Large Scale RNA Production Systems (Promega), following instructions from the manufacturer. The removal of SP6 polymerase was removed through phenol-chloroform extraction, and the RNA replicon was precipitated with 2.5 volumes of 95% ethanol and 0.1 volume of 3M sodium acetate (pH 5.2), according to the manufacturer’s instructions. The synthesized RNA replicons were stored at −80 °C until use. Alternatively, the plasmid TEV-MCS-100A was used to produce the mRNA encoding PfRH5. The RNA generated by the TEV vector did not replicate. We compared the efficiency of these two RNA forms and analyzed their immunological effects.

### 2.3. Liposome Preparation

The liposomes were produced, as previously described [14]. All the lipids were purchased from Avanti Polar Lipids. In order to produce cationic liposomes, DDAB (dimethyldioctadecyl ammonium bromide) lipid vesicles were complexed with DC-Chol, a cationic cholesterol derivative (cholesteryl 3β-N-(di-methyl-amino-ethyl)-carbamate hydrochloride), at a 4:1 molar ratio. The DDAB/DC-Chol lipid vesicles were dissolved in chloroform, and the solution was evaporated under a nitrogen stream to yield a dry lipid film. The remaining chloroform was removed by drying the film under vacuum for 1 h, and the resulting dry lipid film was stored at 4 °C until use. Before use, the liposome preparation was rehydrated in 1 mL of 50 mM HEPES buffer (pH 7.4) to a final concentration of 1 mg/mL, and then maintained at 60 °C under vigorous shaking for 1 h (1000 rpm, Eppendorf tube benchtop mixer). The dispersed liposomes were then sonicated until the suspensions changed from an opaque/milky appearance, which indicated the presence of large particles, to a translucid solution with particles smaller than 100 nm in diameter. A concentration of 8 nM of lipids for each 1 µg of nucleic acid (RNA) was established for the lipoplexes in immunization with sPfRH5 RNA and samPfRH5 replicon. The mixture of liposomes and the nucleic acid of interest was sonicated in an ultrasonic bath for 2 min, and used for transfection and/or immunization immediately after this step.

### 2.4. Zeta Potential and Polydispersity

The charge and size parameters of the formulated liposomes were analyzed using a NanoPlus^TM^ 2 (Zeta Potential Analyzer, Micromeritics, Norcross, GA, USA). The zeta potential describes the electrokinetic potential in colloidal dispersions, thus contributing to the evaluation of charge parameters. Dynamic light scattering (DLS) was used to determine the size (diameter in nm) of the particles in suspension. The polydispersity (PD) of a liposome formulation is a measure of the homogeneity or uniformity of a mixture.

### 2.5. Negative Staining Transmission Electron Microscopy and Cryoelectron Microscopy

For negative staining transmission, the same liposome formulation used for transfection was diluted to a final volume of 1 mL, and a drop of the water-diluted suspension was stained with uranyl acetate for approximately 5 min. The samples were then dried at room temperature and analyzed using a transmission electron microscope (Jeol 100 CX II). For cryoelectron microscopy, 1 mL of plasmid RNA-loaded liposomes was applied over a holey carbon-film grid (Quantifoil Micro Tools, Jena, Germany), followed by flash freezing in liquid ethane using a Gatan Cryoplunge 3 (Gatan, Pleasanton, CA, USA). The visualization of the frozen, hydrated specimen was performed on a JEM2100 electron microscope (JEOL, Tokyo, Japan, operating at 200 kV), resulting in images of 2 μm scale. Images were recorded using a Gatan Ultrascan 4000 CCD camera at 40,000× magnification. Negative staining was performed at the ICB-USP (Institute of Biomedical Sciences, University of São Paulo).

### 2.6. Transfection of the RNA Replicons Stabilized by Liposomes

For transfection, Vero and HEK293T cells were seeded at a density of 10^4^ cells/well, and grown overnight in a 24-well plate. Liposomes at either 4 nM with 2 µg of RNA replicons had their final volumes adjusted with 5% glucose to 50 µL per well, resulting in 1 µg of RNA replicon per mL in each well. Therefore, 50 µL of the formulation was pipetted over the cells, and DMEM medium was added to a final volume of 2 mL per well. After incubation in the presence of liposomes for 24 h, the cells were washed with incomplete RPMI medium, trypsinized, and resuspended in 100 µL of DMEM medium. Half of this volume was analyzed using a flow cytometer (Guava^®^ EasyCyte Mini, Luminex, Austin, TX, USA), in the case of the GFP transfected wells, or using a luminometer (Berthold Lumat LB 9507, Bad Wildbad, Germany), in the case of nanoLuc transfected wells. DMEM was added to the cells, and the plates were incubated. Subsequent analyses were performed 48 h after transfection. The transfection efficiency of the GFP replicons and GFP was evaluated using flow cytometry, with excitation at 488 nm (FL1 filter) to detect the green fluorescent protein (GFP) in the transfected cells. The samples containing nanoLuc were treated with Brefeldin A (BD GolgiPlug™), according to the manufacturer’s instructions, 6–8 h before every analysis. The cells were pelleted at 18,800× *g* for 20 s (Eppendorf centrifuge) and resuspended in Nano-Glo^®^ Luciferase Assay Substrate and Buffer, according to the manufacturer’s instructions, before analysis in the luminometer.

### 2.7. Liposome Toxicity Assay

Vero and HEK293T cells were seeded overnight in 24-well plates at a density of 10^4^ cells/well and transfected with liposomes at various molar ratios (2 nM to 10 nM). After incubation for 24 h, the cells were washed with RPMI incomplete medium, trypsinized, and centrifuged for 5 min at 200× *g*. The pellets were resuspended in working solution from the LIVE/DEAD™ Cell Imaging Kit (Invitrogen, Carlsbad, CA, USA), according to the manufacturer’s instructions, and incubated for 30 min at 25 °C. The samples were then washed twice with phosphate buffered saline (pH 7.4, PBS), and the percentages of live and dead cells were quantified using flow cytometry.

### 2.8. Fluorescence Microscopy of Transfected Cells

Vero cells were grown directly on slides in a Nunc Lab-Tek II Chamber Slide System to a confluence of 10^4^ cells/well. The cells were transfected with liposomes at an 8 nM molar ratio with 6 µg and 2 µg of RNA replicons expressing GFP, and a control well was transfected only with liposomes. Transfection was performed in the same manner as in Section 2.5 in this section, only in a smaller volume (0.5 mL instead of 2 mL per well). After 48 h of transfection, the sections were stained with 0.03 µg/mL DAPI (4′,6-diamidino-2-phenylindole; Sigma-Aldrich, St. Louis, MO, USA), washed with DMEM medium, and the slide was separated for microscopic examination. Figures were acquired using a DMRA2 fluorescence microscope (Leica, Wetzlar, Germany) and MetaMorph software (Molecular Devices Inc., San José, CA, USA), and edited using Figure J (Version 1.51n).

### 2.9. Confocal Immunofluorescence Analysis

Skin sections were embedded in cryoprotectant solution for 48 h, sliced (120 μm) using a Leica CM3050-S cryostat (Leica Biosystems), and mounted on gelatin-coated slides. The sections were then covered with ProLong Antifade reagent (Invitrogen, USA). Images of each section were obtained using a Nikon Eclipse Ti confocal laser microscope (Tokyo, Japan). A laser wavelength of 561 nm was used for GFP detection. The signal analysis was performed using NIS Elements AR4.00.04 (Nikon, Tokyo, Japan) software.

### 2.10. Intradermal Immunization

BALB/c mice of age 5–12 weeks were obtained from the breeding facility for isogenic mice of the Department of Parasitology (ICB/USP), and maintained under pathogen-free conditions during the course of the experiment. Ethical clearance was obtained from the local Ethics Committee at the Institute for Biomedical Sciences/USP (Protocol No. 15/3/3). Groups of five BALB/c mice were sedated with ketamine/xylazine, and had hair at the hindlimb removed using a commercial hair removal cream; the underlying skin was sterilized with 70% ethanol. For the delivery of mRNA/samRNA or DNA as control, 30 µL of liposome/pDNA or RNA mixture at 8 nM concentration was administered in two drops at the hairless skin in the tibia anterior muscle, followed by tattooing of a 2 × 1 cm skin area using a commercial tattoo machine. The tattoo device was adjusted to expose no more than 2 mm of the needle, and was used twice in the muscle for no more than 15 s at a voltage of 16 V set on the power supply. Under these conditions, every mouse received, in the course of one immunization, 10 µg of RNA in 50 µL of liposome solution. Given that some trauma had been caused to the skin, a silicone cream was applied to the tattooed area. Additionally, other groups of mice were immunized with an RNA replicon/liposome mixture. The treated animals were then exposed to an IVIS Spectrum CT (Caliper Life Sciences, Hopkinton, MA, USA) located in the CEFAP-ICB/USP for bioimaging. The tattooed area was analyzed for relative fluorescence (for GFP) or bioluminescence (for nLuc), according to the manufacturer’s protocol. When applicable, euthanasia using a nitrogen chamber was performed in mice to remove treated tissues, which were used for further histological analysis to evaluate the possible effects of the lipoplexes in the dermal tissue and potentially affected cells.

### 2.11. ELISA and Western Blots

Each 96-well ELISA medium binding plate (Jet Biofil, Guangzhou, China) was coated with schizont extract at 200 ng/well and at 4 °C overnight. On the next day, the plates were washed 3 times with PBS/Tween 0.05% (PBS/T) and blocked with 2% skim milk/PBS for at least 1 h at 25 °C. The plates were washed and incubated with antisera generated from mice that were immunized with mRNA and samRNA encoding PfRH5 antigen in a solution of 1% skim milk/PBS. The antisera were endpoint-diluted and incubated for 2 h at room temperature. After four washes with PBS/T, anti-murine IgG coupled to horseradish peroxidase (KPL-Seracare, Milford, MA, USA) was applied for 1 h at 25 °C (1:2500). After repeated washing, all of the wells were developed with TMB substrate (Pierce/Thermo Fisher Scientific), stopped after 5 min with 1 M HCl, and the colorimetric reaction was analyzed using a BioTek plate reader (BioTek, Winooski, VT, USA) at 450 nm/595 nm.

For Western blots, 10 µg of schizont extract or 2 µg of recombinant PfRH5GPI [18] was electrophoresed under non-reducing conditions in SDS-polyacrylamide gels (10%). Afterward, proteins in the gels were transferred onto nitrocellulose membranes (Hybond C; GE Healthcare), and these were blocked with 4% milk in PBS/T for 1 h at room temperature. After four washing steps with PBS/T, the primary antibodies (serum pool of plasmid-liposome immunized animals) were incubated overnight at room temperature at a dilution of 1:500 in 1% milk PBS/T. After five washes with PBS/T, the membranes were incubated with horseradish peroxidase-conjugated anti-mouse secondary antibody diluted 1:3000 for one hour at room temperature. After five washes with PBS/T, the membranes were briefly soaked in ECL reagent (GE Healthcare), and chemiluminescence was documented on Hypermax X-ray films (Kodak) or photographed using a GE Image Quant 3000 apparatus.

### 2.12. P. falciparum Culture and Invasion Inhibition Assays

For the assays with schizont extracts or invasion inhibition, the NF54 strain of *P. falciparum* was maintained in human B^+^ red blood cells (hematocrit 3–5%, ethical clearance for the use of human blood was granted by the local Committee for Ethics in Research involving Humans at ICB/USP, protocol number 803/2016) in RPMI medium supplemented with 0.5% Albumax 1 (Invitrogen/ThermoFisher Scientific). The cultures were kept in airtight boxes (candle jars) at 37 °C, with daily medium changes [19]. Parasitemia was monitored by microscopy using thin blood smears stained with a modified Giemsa stain (Panótico Quick kit; LaborClin, Pinhais, Brazil). The schizont-stage parasites were collected for ELISA and Western blot analyses. The cultures were synchronized with intermittent plasma flotation [20] (Voluven 6%; Fresenius Kabi, Campinas, Brazil), followed by sorbitol lysis [21]. For growth inhibition tests, the cultures were plated in 96-well plates at an initial parasitemia of 1%. Protein A-purified IgG fractions from immunized mice were added at different concentrations (180 μg/mL and 300 μg/mL), and the volumes were matched with RPMI medium for the control IgG (300 µg/mL) from pre-immune sera. Measurements were taken after 24 and 48 h. In order to measure parasitemia, aliquots were removed from wells and stained with ethidium bromide (0.1 ug/mL), and analyzed by flow cytometry (Guava Easycyte Mini), as described previously [22].

### 2.13. Statistical Analyses

All of the experiments were performed at least in triplicate. GraphPad Prism 5.03 (Graph Pad Software, San Diego, CA, USA) and Origin 8 (OriginLab Corporation, Northampton, MA, USA) were used for all of the statistical analyses. Student’s *t*-test was used to compare normally distributed values between groups, and one-way ANOVA or the Kruskal–Wallis Test with Bonferroni or Dunnett’s post hoc test was used to compare three or more groups (*p* < 0.05, considered statistically significant). An analysis using nonlinear regression as dose-response curves was performed to determine the levels of liposome toxicity and compare the transfection efficiency between groups.

## 3. Results

### 3.1. Characteristics of DDAB/DC-Chol Liposomes and Liposomes/RNA Replicon Complexes

DDAB-Chol liposomes containing the RNA replicons (1 μg of samRNA) containing different amounts of encapsulating cationic lipids were characterized by measuring the zeta potential, polydispersity index and diameter, using a dynamic light scattering analyzer (Table 1).

The zeta potential or electrokinetic potential of liposomes is a measure of their charge. Therefore, a positive value is expected for cationic liposomes (containing the cationic lipid DC-CHOL), provided that it represents the overall charge of liposomes in solution. This was also true for liposomes that were not complexed with nucleic acids (Table 1). The presence of RNA replicons in this formulation is a relevant factor for altering the negative charges intrinsic to the nucleic acid molecules. This is due to the molar ratios of liposomes of 2 nM to 8 nM, where the RNA replicon does not appear to be completely encapsulated, yielding a negative charge. When using higher concentrations of liposomes until complete encapsulation of the replicons, the predominance of a positive charge is observed (≥16 nM). Notably, average zeta potentials of −9 to −23 mV were also reported in studies that used complexes of cationic liposomes to deliver plasmid DNA in vitro [14], underscoring the selection of molar ratios of 4 nM and 8 nM to conduct the in vitro and in vivo assays. A polydispersity index (PDI) was used to determine the particle size distribution. A PDI < 0.2 indicates that the sample is monodisperse, while a PDI > 0.7 indicates that the size distribution is very broad. Our PDIs were close to 0.2, which was consistent with the monodisperse particle size distribution. The diameter of the liposomes was expected to be less than 100 nm, as seen later in the NS-TEM (Figure 2), which was consistently observed in the 4 nM and 8 nM samples. As the concentration of liposomes increased, so did their size. Some of the samples did not present consistent measurements for diameters greater than 1000 nm. These were likely aggregates of liposomes (also shown in Figure 2) that formed spontaneously. These initial analyses established the working range of the RNA replicon: the liposome ratio used in subsequent experiments.

### 3.2. Morphology Analysis of DDAB/DC-Chol Liposomes and Lipoplexes

Liposomes containing the RNA replicon (1 µg of RNA/8 nM of lipids) expressing GFP were analyzed using negative-stained transmission electron microscopy (see Figure 2). The morphology and architecture of the liposomes were evaluated, and particles smaller than 100 nm were identified, as expected for sonicated lipoplexes.

The negative staining procedure generates a contrast between the liposomes and the electron-dense material in which they are embedded. In Figure 2, light-colored, round liposomes with darker colors are visible. The population of liposomes identified seemed heterogeneous in size and shape; some large unilamellar vesicles (LUVs) between 50 and 500 nm and some small unilamellar vesicles (SUVs) under 50 nm (Figure 2B) were considered the ideal size for this study. Some artifacts were also present, probably due to the staining process. Distortions of the round shape of the vesicles were due to the drying steps, which were necessary when preparing the sample, and the staining process. The aggregation of liposomes occurred over time, as shown in Figure 2A; around isolated populations of SUVs instead of a dark background, there was a bright lipid aggregate, which could be broken down into smaller vesicles again by sonication [23].

### 3.3. Fluorescence Microscopy Reveals Transgene Expression of samRNA-Transfected Vero Cells

In vivo transfection of mammalian cells with GFP-encoding samRNA/liposome complexes was visualized via fluorescence microscopy. Vero cells transfected with cationic liposomes were analyzed 48 h post-transfection using a fluorescence microscope. Different amounts of the replicon RNA were evaluated: transfection with 6 µg of replicon, 2 µg of replicon, and a control without the use of replicons (Figure 3).

This qualitative analysis indicated that the RNA replicon was taken up and translated by transfected cells. Additionally, an increase in the number of replicons led to an increase in GFP expression. Cells transfected only with liposomes did not show the same fluorescence pattern as cells transfected with replicons, indicating that the observed fluorescence did not arise from the lipid vesicles.

### 3.4. Flow Cytometry Analyses for Determination of GFP Expression and Toxicity of samRNA Combined with Different Quantities of Cationic Liposomes

For their use in vaccines, cationic liposomes must deliver RNA replicons encoding a gene of interest with integrity and efficiency, so that they can be translated into antigens of interest for processing and presentation. To assess the efficiency of replicon transfection, assays were performed with GFP-expressing replicons, which were first evaluated using flow cytometry (Figure 4).

GFP-expressing replicons were encapsulated in cationic liposomes that were transfected into Vero or HEK293T cells, and analyzed 24 and 48 h after transfection. The fluorescence levels were compared between non-transfected samples and samples transfected only with liposomes (both controls exhibited very similar levels of fluorescence, as shown in Figure 4A,C). When quantifying the results via flow cytometry analysis for protein expression, ~45.7% of the total cells transfected expressed GFP 24 h after transfection, which was reduced by ~40.2% 48 h after transfection (refer to histograms in Figure 4C), but still presented a robust GFP signal. The threshold between fluorescent-positive or fluorescent-negative cells was set such that >99% of non-transfected Vero cells were considered fluorescent-negative. The scatter plot in Figure 4B clearly distinguishes the population of cells exhibiting fluorescence from cells without any fluorescence. After detecting robust signals of GFP expression in Vero cells, assays were conducted to evaluate the influence of the quantity of inoculated RNA and the number of liposomes used for transfection. In the HEK239T cells, an average of ~67.3% were expressing GFP (Figure 4D,E), but the fluorescence signal was not as strong as that in Vero cells (Figure 4A–C). When analyzing the effect of different replicon quantities, GFP fluorescence increased in proportion to higher concentrations of transfected replicons. This result was confirmed using fluorescence microscopy in Figure 2. This was also represented by a shift to the right in the population of fluorescent cells, indicating higher levels of fluorescence as detected by flow cytometry (Figure 4C). In this case, by increasing the quantity of replicons, the amount of liposomes also increased (1 μg of replicon to 10 μL of liposomes) to maintain the same proportion and reduce the influence of liposomes in the delivery, thus exclusively focusing on the amount of replicon. The same procedure was applied for transfection with increasing concentrations of liposomes, maintaining a constant amount of transfected replicon (Figure 4D), showing that increasing the concentration of liposomes increased the delivery of RNA replicons.

The cytotoxic effects of different concentrations of DDAB-Chol liposomes were investigated, keeping in mind that these cationic particles are known to have low toxicity and low immunogenicity. The percentage of live cells was quantified using flow cytometry (Appendix A) by counting cells negative for the cell imaging stain (LIVE/DEAD™ Cell Imaging Kit, Invitrogen). Our liposome formulations did indeed show cytotoxicity, and the results clearly showed that as liposome concentrations increased, cytotoxicity increased. High concentrations of liposomes (10 nM) induced up to 27.9% cytotoxicity after 24 h of incubation, comparable to what was shown by others [13]. These liposome compositions still presented lower toxic effects on Vero cell viability than other polycations, e.g., polyethyleneimine was reported to induce 50% cytotoxicity within an hour of incubation [24]. The best combination of low toxicity and high expression levels was used in the subsequent experimental immunization by tattooing and cell transfection.

### 3.5. Detection of GFP Protein and Luciferase Luminescence Detection in Vero Cells

To further verify the quantity of transgene expression, Vero cell extracts from the transfection assays (Figure 5A) were analyzed qualitatively under UV light. To quantify transgene expression, Vero cells were transfected, with or without nLuc replicons, as described previously, centrifuged and lysed with cell lysis buffer. The lysates were resuspended in Nano-Glo^®^ Luciferase assay substrate, and their luminescence activity was analyzed in a luminometer. The transfected cells produced a considerably stronger signal, which was distinguishable from that of the non-transfected control (Figure 5B,C). The luminescence signal in the transfected cells was four times higher than that in the control cells.

### 3.6. Confocal Microscopy Imaging from Tissue Tattooed with samGFP RNA Show GFP Expression

To analyze the effect in vivo of the tattooing strategy for samRNA immunization, BALB/c mice were tattooed with liposome-packaged replicon RNA encoding the GFP protein. The effect of the intradermal inoculation of samGFP RNA was easily observed as light-green spots in the immunization scar upon exposure to normal or invasive black light (Figure 6A). In order to compare the efficiency of samRNA with plasmid DNA intradermal inoculation, we compared 10 µg of a plasmid encoding GFP and samRNA encoding GFP. Dermal inoculation resulted in an increase in GFP production in both cases, with a larger number of cells and a higher fluorescence in animals treated with samRNA (Figure 6B) when analyzing confocal sections from the dermis and epidermis treated by tattooing. These findings suggest that intradermal immunization with samRNA and conventional plasmid DNA can be used for dermal deposition, with possible immune effects.

### 3.7. Immunization by Tattooing samRNA and TEV mRNA-Encoding PfRH5 Antigen Elicits an Antibody Response against the Native and Recombinant Antigen

Next, we investigated whether the samRNA and mRNA encoding the promising malaria vaccine candidate, PfRH5, led to a humoral immune response after immunization by tattooing. The samRNA- or TEV-mRNA-encoding PfRH5 was administered intradermally (10 µg each, loaded in cationic liposomes, two doses in two-week intervals). One week after the last RNA inoculation, sera from the animals were collected and analyzed for their anti-PfRH5 content via endpoint dilution ELISAs using a recombinant form of the protein. Both replicon- and TEV-RNA-encoded PfRH5 elicited antibodies with titers as high as 1:12,000. To verify which antigen was recognized, the sera were also tested with Western blotting against trophozoite/schizont extracts (Figure 7B), which showed that an antigen the size of PfRH5 was recognized.

Finally, the anti-malarial growth-inhibitory effects of the sera generated by tattoo immunization were tested. For this, purified IgG fractions were applied in *P. falciparum* NF54 cultures, and the growth of parasites was compared to cultures treated with 300 μg/mL IgG from pre-immune animals. As shown in Figure 7C, both the samRNA and TEV-mRNA constructs decreased the parasite growth in a dose-dependent manner. The only notable difference was that the antibodies created by TEV RNA failed to interfere with parasite growth at 48 h, whereas antibodies from replicon-immunized animals still exerted an inhibitory effect, although to a lower degree. The shown values were calculated by comparing parasitemia in parallel cultures supplemented with 300 µg/mL non-related IgG (=0% growth inhibition, for samPfRH5 is shown to average 23% ± 1.5% for 180 µg/mL and 42% ± 2.4% for 300 µg/mL in 24 h; average 14.5% ± 4.5% for 180 µg/mL and 25% ± 9.5% for 300 µg/mL in 48 h; in the case of TEV PfRH5, average of 29.4% ± 2% for 180 µg/mL and 49% ± 3.5% for 300 µg/mL). The analyses assuming equal variance and non-parametric values showed no difference between 24 h for both RNA, and a large difference after 48 h of cultivation (Kruskal–Wallis Test, *** = 0.0001). These results indicate the possibility of using tattooing immunization with cationic lipoplexes based on the RNA-encoding antigens of *P. falciparum*.

## 4. Discussion

The feasibility of using mRNA as an immunogen has long been discussed, and mRNA immunization has become pivotal in combating the SARS-CoV-2 pandemic. A breakthrough in mRNA immunization became possible after the discovery that nucleotides in the delivered mRNA must be modified to avoid unwanted inflammatory responses against the RNA itself [25]. Since the production of synthetic mRNA and encapsulation does not depend on the sequence of the genes encoded, the combination of modified nucleotides and self-amplifying RNA, such as from the engineered alphavirus genomes used here, theoretically represents a platform for a swift response to rapidly emerging global health threats, such as the SARS-CoV-2 pandemic. In this study, we demonstrated the feasibility of samRNA vaccines packaged in cationic liposomes delivered intradermally with a tattooing device by detecting GFP expression in tattooed skin and generating anti-*P. falciparum* inhibitory antibodies against the blood stage antigen PfRH5.

Skin tattooing is an intradermal immunization approach that has been applied to DNA vaccinations to induce stronger immune responses than intramuscular needle injections combined with adjuvants [26]. Some key advantages make skin tattooing a good choice for intradermal immunization: (i) it covers a large skin area, which could potentially elicit a stronger immune response, bearing in mind that it is a region abundant in antigen-presenting cells; (ii) it is inexpensive; (iii) it produces low DNA damage and (iv) it recruits immune cells, such as macrophages and dendritic cells, enhancing the immune process [27]. The level of expression suggests that tattooing RNA robustly promotes antigen production in the dermis. Surprisingly, tattoo-transfection of RNA showed a stronger induction of reporter proteins than DNA transfection (see Figure 6C). In contrast to other groups that attempted to decrease the inflammatory effects of unmodified RNA during immunization [25], a large production of GFP was observed following the delivery of unmodified samRNA. Although a large amount of RNA (10 µg) was initially used, it can be assumed that only a small portion was effectively transferred to the skin. This probably occurred because a conventional tattooing machine was used, in contrast to previous studies in which specific multilayer tattooing delivery was more effective [28]. Notably, the mRNA and samRNA used in this study were not produced using a capping reagent or pseudo-uridine, which can improve the translation of RNA molecules [25]. This highlights the efficiency of the method applied, but also simplifies the application of RNA transfection or immunization when sophisticated devices are not available. Additionally, the advantage of samRNA is that it can at least partially overcome the suppression of the foreign unmodified RNA immunization effect because the self-replicating RNA inside the transfected cell is modified by the cell itself.

Liposomes have advantages over other drug delivery systems because of their ability to solubilize hydrophilic molecules in the aqueous phase, preventing their degradation and precise release until the target is reached. They can also carry lipophilic molecules attached to their lipid bilayers when drugs show low solubility in water. Liposomes can be classified according to their size and number of lipid bilayers. While small unilamellar vesicles (SUVs) are typically smaller than 50 nm, large unilamellar vesicles (LUVs) range between 50 and 500 nm in size, and giant unilamellar vesicles (GUVs) are capable of reaching up to 100 µm [29]. The particle size is especially important when designing synthetic vaccines because it affects their dispersion and diffusion speeds, drainage to the lymph nodes, and antigen concentration [30]; these ultimately regulate intracellular uptake by antigen-presenting cells. In the present study, the encapsulation method was tested with the successful incorporation of a DNA vaccine encoding PfRH5 [31]. As performed previously with plasmid DNA, tests to determine the ideal lipid/cargo ratio were conducted, and it was possible to obtain cationic particles with an ideal size of 50–100 nm, which protected their cargo and were easily taken up by antigen-presenting cells [32].

The obtained antiPfRH5 titers and their inhibitory effects were perfectly comparable in efficiency to previous results obtained by us [31] and others [33]; we are currently not aware of any other published study that employed samRNA immunization of PfRH5. In terms of inhibitory activity, the samRNA performed less well than when native proteins were incorporated into liposomes [34,35], but performed better than liposomes with recombinant PfRH5 displayed on their surface [18]. This may be because merozoite protein-loaded particles elicit antibodies against multiple antigens, including antiPfRH5, which then function synergistically to slow down erythrocyte invasion [36]. It must be noted that the non-polymorphic antigen PfRH5 alone is not sufficiently effective as an anti-blood stage vaccine, because only specific inhibitory antibodies against determined epitopes provide protection [37]; even when delivered in PfRH5′s natural complex with CYRPA and RIPR, invasion inhibition is not greatly augmented [38], indicating that more antigens are needed to block merozoite invasion. In this sense, samRNA encoding a number of different antigens against which antibodies exert invasion-inhibitory activity, seems to be an attractive avenue to follow in terms of vaccine development. An additional advantage of mRNA-encoded antigens lies in the stimulation of cellular immune responses, as proteins synthesized after translation from delivered mRNA are internally processed and presented as MHC1 molecules, thus calling for the inclusion of liver-stage antigens in a samRNA-based approach.

## Figures and Tables

**Figure 1 pharmaceutics-15-01223-f001:**
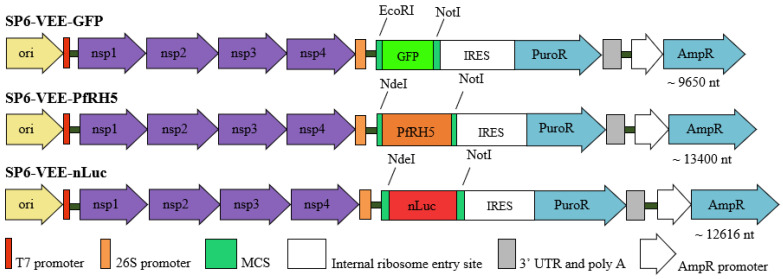
Plasmid constructs for RNA replicon synthesis. The proteins nsp1, nsp2, nsp3 and nsp4, after the origin of replication (ori), are nonstructural alphavirus proteins, while the other proteins represented are transgenes (GFP and nLuc). The nLuc and PfRH5-amplified PCR products were cloned in the multiple cloning site (MCS) of the basic replicon vector. GFP and nanoluciferase were reporters used to monitor the transfection efficiency. The GFP protein was located in the cytosol, and was detected in transfected cells using flow cytometry and immunofluorescence of the transfected tissue. The nanoluciferase protein was secreted from the cells using the tissue plasminogen activator (tPA) signal sequence, and its bioluminescence could be measured using a luminometer (in vitro assays), and also through an in vivo imaging system (IVIS^®^ Spectrum). The plasmids also included puromycin and ampicillin resistance genes (PuroR and AmpR).

**Figure 2 pharmaceutics-15-01223-f002:**
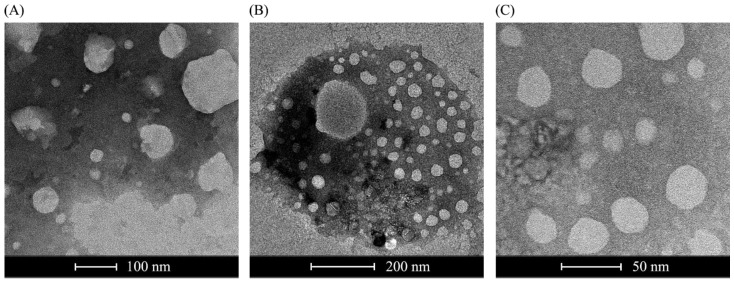
NS-TEM (negative-stained transmission electron microscopy) images of the formulated liposomes showing the final size of packaged replicons/samRNAs. (**A**) Cationic liposomes alone; (**B**) cationic liposomes complexed with 1 μg of GFP replicon; and (**C**) cationic liposomes complexed with 1 μg of nLuc replicon. Data scales of 100 nm, 200 nm and 50 nm, respectively.

**Figure 3 pharmaceutics-15-01223-f003:**
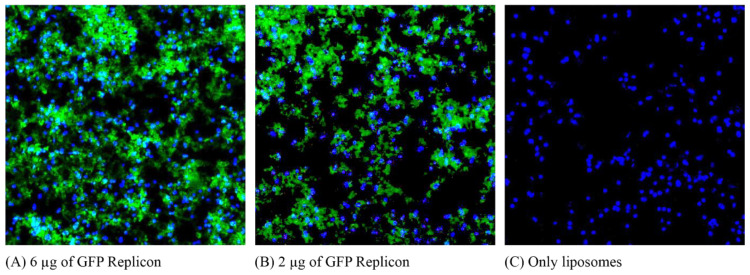
Vero cells were easily transfected with packaged samRNA encoding GFP. Comparison of Vero cells transfected with (**A**) 8 nM of cationic liposomes conjugated with 6 µg of samGFP RNA; (**B**) 8 nM of cationic liposomes conjugated with 2 µg of samGFP; (**C**) only liposomes. The nuclei were stained with DAPI, and the green cytoplasm of the transfected cells indicates GFP expression. Images were captured at 40× magnification.

**Figure 4 pharmaceutics-15-01223-f004:**
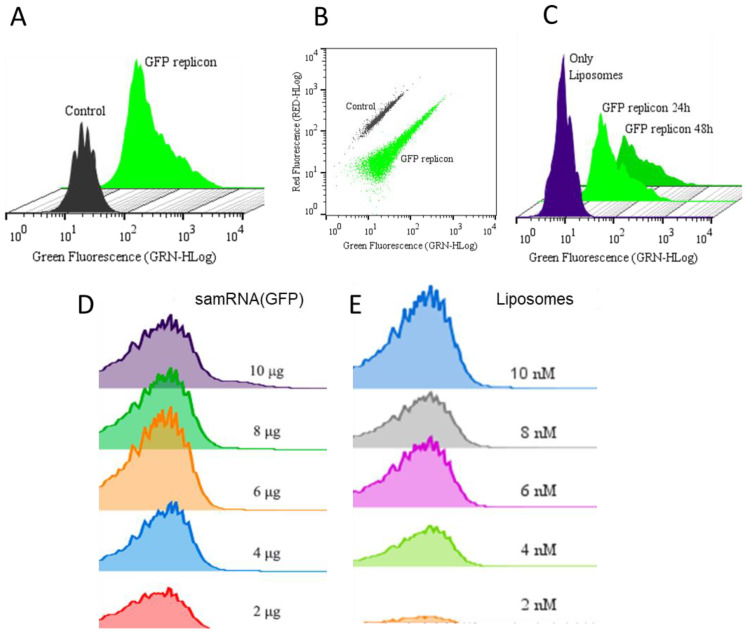
Comparison of samRNA or lipid composition quantities in transgene expression in vitro. (**A**) Representative histograms of Vero cells expressing GFP 24 h after transfection with the GFP-expressing replicon in comparison to an untransfected control. (**B**) Dot plot comparing the populations described in (**A**). (**C**) Representative histograms of GFP expression 24 h and 48 h after transfection with samRNA encoding GFP encapsulated in liposomes in comparison to a control transfected with empty liposomes. (**D**) Representative histograms of GFP expression dependent on the amount of transfected replicon (2 μg, 4 μg, 6 μg, 8 μg or 10 μg of transfected replicons) in HEK293T cells 24 h after transfection. (**E**) Representative histograms of GFP expression in dependence on the concentration of liposome used (2 nM, 4 nM, 6 nM, 8 nM or 10 nM) to deliver 1 μg of GFP replicon to HEK293T cells 24 h after transfection.

**Figure 5 pharmaceutics-15-01223-f005:**
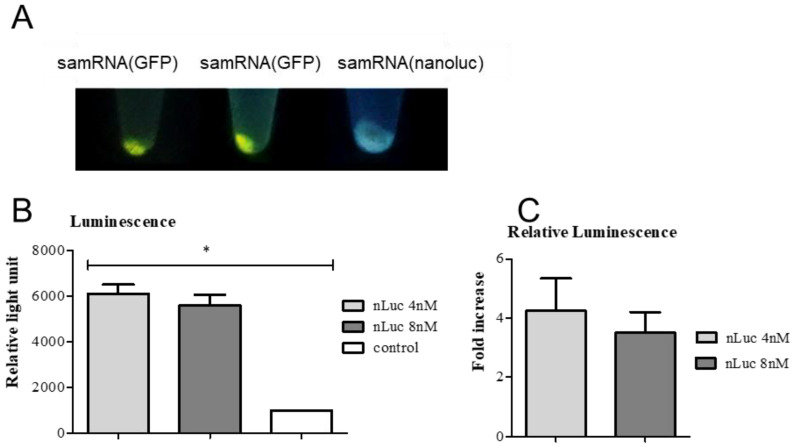
Quantification of GFP and Nano-Luciferase expression demonstrated their strong production in Vero cells. In (**A**), pellets from Vero cells transfected with replicons expressing either GFP or nanoluc are shown under UV light (48 h after transfection). The third sample was transfected with cationic liposomes containing nLuc replicons under the same conditions. (**B**) Luminescence activity of Vero cells transfected with nLuc replicon complexed with 4 nM or 8 nM of cationic liposomes compared with non-transfected cells (control). The luminescence from each sample was normalized regarding the volume of the sample and protein concentration from each one. *p*-values are given for one-way ANOVA where * represents *p* < 0.005. (**C**) Relative luminescence of Vero cells transfected with nLuc replicon in relation to non-transfected cells. The difference in luminescence between the two samples (with a higher and lower concentration of liposomes) was not statistically significant. All experiments were performed in triplicate. Error bars show the deviations from three technical replicates.

**Figure 6 pharmaceutics-15-01223-f006:**
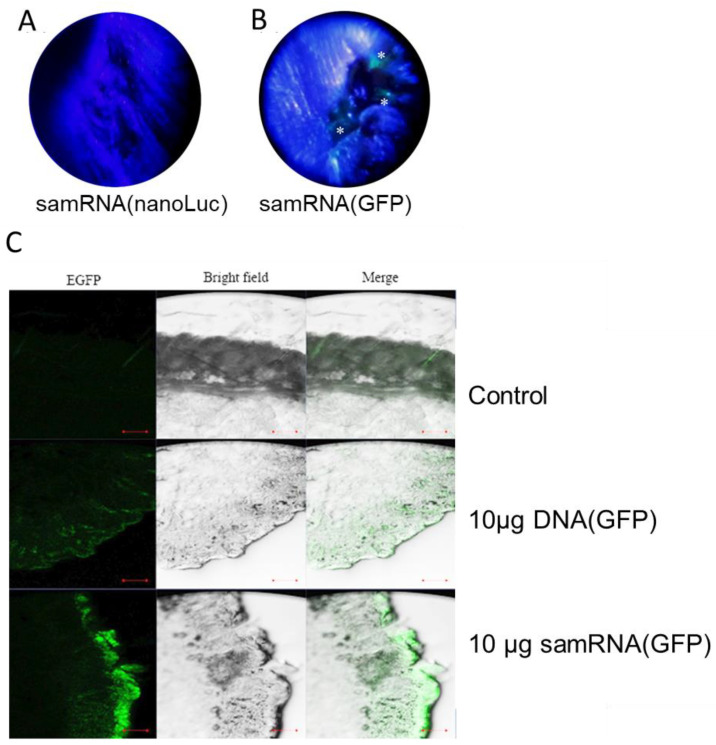
Stronger GFP expression was observed after tattooing samRNA compared with plasmid DNA. Comparison of tattoo scars of animals immunized with nLuc expressing replicon (**A**, negative control) and GFP-expressing replicon (**B**), respectively, in cationic liposomes, observed under black light. Asterisks depict green spots on the skin of the animal tattooed with GFP replicons 48 h after the intradermal inoculation. Histological analyses of tissue sections (**C**) of BALB/c mice tattooed with liposomes encapsulating either 10 µg of DNA plasmid or RNA-replicon-encoding GFP 48 h after inoculation in the dermis. Control mice were not tattooed, and fluorescence in the control mice is due to hair in the tissue section. Images were acquired by confocal laser scanning microscopy. Data scales of 100 µm.

**Figure 7 pharmaceutics-15-01223-f007:**
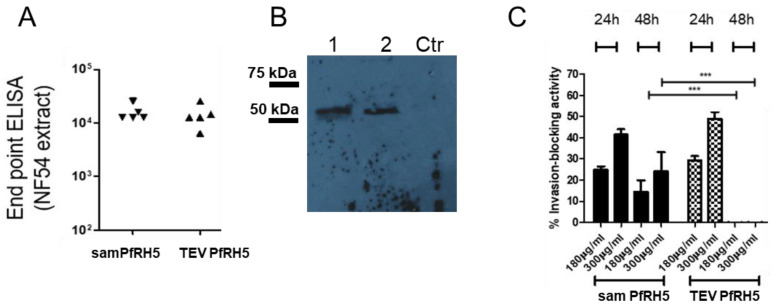
Sera from immunized mice recognized native PfRH5 in *P. falciparum* NF54 schizont extracts, and exerted an inhibitory effect in vitro. In (**A**), endpoint titers of anti-samPfRH5 and TEV mRNA PfRH5 sera were measured (titers of Log10, no statistical difference between the groups). Individual sera of 5 liposome-packaged RNA-immunized mice were tested in endpoint dilutions. For coating, ELISA plate schizont extracts were used instead of recombinant PfRH5 antigens. (**B**) Sera from samPfRH5- and TEV PfRH5 RNA-immunized mice were used to recognize extracts from *P. falciparum* NF54. Lane 1 is pooled sera from samPfRH5-immunized mice, lane 2 from TEV PfRH5-immunized mice, and Ctr, controls from pre-immune sera. In (**C**), the percentage of growth inhibition of parasite proliferation (measured by flow cytometry) is shown for two dilutions of purified IgGs from four mice immunized with encapsulated samPfRH5 or TEV PfRH5. The shown values were calculated by comparing parasitemias in parallel cultures supplemented with 300 µg/mL non-related IgG (=0% growth inhibition, for samPfRH5 is shown to average 26% ± 1.5 for 180 µg/mL and 43.1% ± 2.4 for 300 µg/mL in 24 h; average 214.5% ± 4.5 for 180 µg/mL and 24% ± 9.5 for 300 µg/mL in 48 h; in the case of TEV PfRH5, average of 29.4% ± 2 for 180 µg/mL and 49% ± 3.5 for 300 µg/mL). Analyses assuming equal variance and non-parametric values showed no difference between 24 h antisera from both RNA immunizations, and a large difference after 48 h of cultivation, Kruskal–Wallis test, *** = 0.0001) The values are shown for 24 h and 48 h of inhibition. See Appendix A for original Western blot X-ray film.

**Table 1 pharmaceutics-15-01223-t001:** Zeta Potential, polydispersity and diameter of formulated liposomes at different molar ratios. Measurements were performed in triplicate with cationic liposomes alone or encapsulating 1 μg of RNA replicons encoding GFP with increasing concentrations of liposomes. Unstable ranges of measurements are marked with *.

Formulation	Liposomes	Liposomes + RNA Replicon-Encoding GFP ^#^				
		2 nM	4 nM	8 nM	16 nM	32 nM
Zeta Potential (mV) Average ± SD	15.15 ± 1.18	−6.01 ± 0.48	−9.57 ± 0.07	−22.95 ± 1.55	4.4 ± 0.05	8.9 ± 0.2
Polydispersity index Average ± SD	0.225 ± 0.02	0.262 ± 0.01	0.212 ± 0.02	0.221 ± 0.01	*	*
Diameter (nm) Average ± SD	*	*	43.17 ± 1.62	89.94 ± 0.95	155.5 ± 2.05	*

^#^ All measurements were carried out in triplicate with cationic lipid liposomes alone or with incorporation of the same amount (1 μg) of RNA replicons. * Unstable range of measurements.

## Data Availability

All data are contained in the article body/figures.

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
