# Peer review of "Establishment of an Antiplasmodial Vaccine Based on PfRH5-Encoding RNA Replicons Stabilized by Cationic Liposomes"

_pharmaceutics, 2023, doi:10.3390/pharmaceutics15041223_

Round 1
Reviewer 1 Report
The article is well-written and includes all the essential details to support the study outcomes. There were a few typographic and grammatical errors areas as I highlighted in the attached manuscript. These errors are related to the scientific name and references.
- In reference 8 article is missing the author's details and journal name (PMID: 2011582).
- Please clarify the DAPI concentration, is this 0.03μL/mL or 0.03μg/mL?
- In figure 5 panel name A and B confuses lane A and B, please modify that for more clarity.
- It would be great if you could provide a better western image in figure 5.

Author Response
Reviewer 1
The article is well-written and includes all the essential details to support the study outcomes. There were a few typographic and grammatical errors areas as I highlighted in the attached manuscript. These errors are related to the scientific name and references.
- In reference 8 article is missing the author's details and journal name (PMID: 2011582).
The bibliography list was fully revised and references with missing information have been corrected in the revised version.
- Please clarify the DAPI concentration, is this 0.03μL/mL or 0.03μg/mL?
The DAPI concentration is indeed 0.03 μg/ml. We corrected this accordingly.
- In figure 5 panel name A and B confuses lane A and B, please modify that for more clarity.
We have changed the figure accordingly.
- It would be great if you could provide a better western image in figure 5.
We agree that the figure is of bad quality and prefer to delete it. Since nLuc expression can be quantified directly without antibodies, we believe that the nLuc experiments shown in this figure are sufficient to show the performance of the transfection.
Reviewer 2 Report
The manuscript is well written, I don't have any comments
Author Response
Thank you very much for your evaluation. We have thoroughly revised the text for English language mistakes.
Reviewer 3 Report
Wesley L. Fotoran and co-authors have presented the study based on the design of cationic lipids to carry RNA. The manuscript is well organized and covers all the major aspects. Some queries are listed below:
1. The abstract should be rewritten; some major information should be included in the introduction part along with the references.
2. The role of liposomes (anionic or cationic) should be explained in more detail in the introduction part.
3. The scope of the lipid selection is very narrow, so pre-screening for a set of lipids should be done.
4. The higher zeta potential value up to +60 mV is easy to attain, and so many reports can be found on the literature. Why the author selected only this lipid composition
5. The cytotoxicity should also be considered.
Good luck
Author Response
Reviewer 3
Wesley L. Fotoran and co-authors have presented the study based on the design of cationic lipids to carry RNA. The manuscript is well organized and covers all the major aspects. Some queries are listed below:
- The abstract should be rewritten; some major information should be included in the introduction part along with the references.
Both abstract and introduction have been rewritten in order to add more information and to better explain the relevant points.
- The role of liposomes (anionic or cationic) should be explained in more detail in the introduction part.
We have added two sentences and a new reference (review) in lines 67ff.
- The scope of the lipid selection is very narrow, so pre-screening for a set of lipids should be done.
Our group has previously worked on delivery of DNA in liposomes (references 16 and 31) and found out that the here used lipid composition was very suitable for the packaging of RNAs.
- The higher zeta potential value up to +60 mV is easy to attain, and so many reports can be found on the literature. Why the author selected only this lipid composition?
As described above, the herein used lipid composition proved efficient for the purpose. However, it is not excluded that other lipid compositions may work equally or better. We have used a lipid composition with the lowest toxicity and the highest transfection efficiency.
- The cytotoxicity should also be considered.
This is an important point and we are grateful for the assessor to emphasize the issue. We now show the results of experiments regarding toxicity and briefly comment on them (chapter 2.7 lines 183-191, results: 282-283, discussion 386-398, Supplementary Figure 1, lines 733ff.)
Round 2
Reviewer 3 Report
All queries have been fully satisfied.